# Rheumatoid Arthritis in the View of Osteoimmunology

**DOI:** 10.3390/biom11010048

**Published:** 2020-12-31

**Authors:** Mélanie Auréal, Irma Machuca-Gayet, Fabienne Coury

**Affiliations:** 1INSERM UMR1033 LYOS, University of Lyon I, 69003 Lyon, France; melanie.aureal@chu-lyon.fr (M.A.); irma.machuca-gayet@inserm.fr (I.M.-G.); 2Department of Rheumatology, Lyon Sud Hospital, 69310 Pierre-Bénite, France

**Keywords:** rheumatoid arthritis, bone erosion, inflammatory bone loss, osteoclast

## Abstract

Rheumatoid arthritis is characterized by synovial inflammation and irreversible bone erosions, both highlighting the immense reciprocal relationship between the immune and bone systems, designed osteoimmunology two decades ago. Osteoclast-mediated resorption at the interface between synovium and bone is responsible for the articular bone erosions. The main triggers of this local bone resorption are autoantibodies directed against citrullinated proteins, as well as pro-inflammatory cytokines and the receptor activator of nuclear factor-κB ligand, that regulate both the formation and activity of the osteoclast, as well as immune cell functions. In addition, local bone loss is due to the suppression of osteoblast-mediated bone formation and repair by inflammatory cytokines. Similarly, inflammation affects systemic bone remodeling in rheumatoid arthritis with the net increase in bone resorption, leading to systemic osteoporosis. This review summarizes the substantial progress that has been made in understanding the pathophysiology of systemic and local bone loss in rheumatoid arthritis.

## 1. Introduction

Rheumatoid arthritis (RA) is a chronic systemic autoimmune disorder that primarily causes tenderness, swelling, and destruction in joints with the resulting disability. While its precise etiology is unknown, RA is considered to develop as a result of interactions between inherited genetic factors and environmental triggers during pre-clinical phases of the disease, in which tolerance is broken and autoantibodies, including anti-citrullinated peptide antibodies (ACPAs), are produced long before the appearance of the first joint symptoms. RA is characterized by dysregulated inflammatory and immune processes in the synovium of the joints as well as bone loss, highlighting this disease as an excellent model for gaining insights into osteoimmunology.

Immunology and bone biology have been separately studied for a long time despite the fact that close relationships have long been noted with pioneering works in the early 1970s [1,2]. The term osteoimmunology was first coined by Aaron and Choi in 2000 to define the crosstalk between the immune and bone systems [3]. Many factors classically considered immune-related, such as cytokines and transcription factors, are actually shared by both systems.

Herein, we highlight the key concepts and recent advances in the osteoimmunology field within the context of RA.

## 2. Bone Loss in RA

The bone is a dynamic tissue that continuously remodels to maintain skeletal integrity through a balance between the activity of osteoclasts and osteoblasts, which resorb and synthesize the bone, respectively. Under physiological conditions, the activity of the osteoclasts and osteoblasts is tightly controlled, such that the amount of bone which is removed is exactly replaced. The tight interaction between the immune system and bone is illustrated in RA by chronic inflammation that disrupts this physiologic balance, favoring bone resorption over bone formation and leading to bone loss in the absence of ectopic bone formation in contrast to spondyloarthritis. Bone loss is an early and common feature in RA and occurs in three forms during the course of the disease: Localized bone loss in the form of articular bone erosions, periarticular osteopenia adjacent to inflamed joints, and generalized bone loss involving the axial and appendicular skeleton.

Although articular bone erosions can be observed in many rheumatic diseases, they have been included in diagnostic criteria for RA as they are commonly considered to be the hallmark of the disease [4]. Neatly demarcated and located at the subchondral bone and the articular margins where the inflamed synovial tissue comes into direct contact with the bone, bone erosions in RA are observed predominantly in metacarpophalangeal, proximal interphalangeal, and metatarsophalangeal joints (Figure 1) [5]. Articular bone erosions first affect the cortical bone surface and then often the adjacent trabecular bone. Once formed, they are considered to not repair. The imaging evaluation of bone erosions is used for the diagnosis and prognosis of RA, as well as for monitoring the efficacy of disease-modifying antirheumatic drugs (DMARDs) in retarding or even stopping bone erosion occurrence in daily clinical practice and in all major clinical trials. Conventional radiography is the mainstay of bone erosion detection, but ultrasonography and magnetic resonance imaging (MRI) have emerged as more sensitive methods than radiography [6]. Bone erosions are a key outcome in RA and correlate with disease severity and functional deterioration. Erosions emerge early in RA, even prior to the clinical onset of arthritis in still asymptomatic ACPA-positive subjects [7]. Joint damage progression is also most rapid within the first years of disease. For instance, almost half of the patients with early RA (mean disease duration of 1 year) had erosions at inclusion and 90% became erosive after 2 years in a study over a period of 10 years [8].

Periarticular osteopenia affects the trabecular bone adjacent to inflamed joints and is one of the imaging hallmarks of RA, along with bone erosions and uniform joint space narrowing. Indeed, bone mineralization is usually normal in all arthropathies except RA. Periarticular demineralization is also the earliest morphological feature of RA, preceding bone erosion and joint space narrowing [9]. However, although the 2010 American College of Rheumatology/European League Against Rheumatism classification criteria for RA are expected to identify RA at an early stage, radiographic periarticular osteopenia has not been included in these criteria [4]. This is due to the fact that it is challenging to detect the periarticular density changes in the early stage of the disease by conventional radiography and the development of more sensitive and reproducible imaging analysis for detection is needed.

Generalized bone loss involving the axial and appendicular skeleton is an important co-morbidity in RA. Patients with RA across all age groups and regardless of sex or anatomical sites are at an increased risk of osteoporosis and osteoporotic vertebral and non-vertebral fractures as compared to healthy patients [10,11]. Apart from inflammation, many factors can influence the generalized balance between bone resorption and bone formation, such as glucocorticosteroids, used in the treatment of RA, and immobility due to pain and swelling of the joints, as well as muscle weakness [12,13]. In addition, RA more frequently affects women in the perimenopause and is often associated with smoking, both conditions which lead to osteoporosis.

## 3. Increased Bone Resorption in RA

### 3.1. Osteoclasts Mediate Bone Destruction in RA

Osteoclasts are the only cell type capable of bone resorption. To resorb the bone, osteoclasts secrete protons, which acidify the extracellular compartment beneath their ruffled border in order to solubilize calcium phosphate, and proteases, such as cathepsin-K, which degrade the exposed organic matrix of bone [14]. Significant evidence has accumulated demonstrating the crucial role of osteoclasts in arthritic bone degradation. Indeed, multinucleated cells exhibiting an osteoclastic phenotype, which includes tartrate-resistant acid phosphatase, cathepsin-K, and the calcitonin receptor, have been identified at sites of bone erosion at the pannus-bone interface both in RA patients [15,16] and animal models of arthritis [17,18,19,20,21]. This role was finally demonstrated by osteoclast-deficient mouse models of arthritis, which were shown to be fully protected from bone erosion [20,21]. In the first study, osteopetrotic mice, in which osteoclastogenesis is impaired, were used as recipients of K/BxN serum transfer arthritis. In the second study, mice lacking c-fos, a transcription factor essential for osteoclastogenesis, were crossed with transgenic mice that over-express human TNF (hTNFtg). In both cases, mice were fully protected from erosion despite synovial inflammation.

Enhancement of the osteoclast function is the key component of bone degradation associated with RA. Increased osteoclast activity results from a composite action: An increase in their activation, their lifespan and their number, the latter due to an enhanced recruitment of osteoclast precursors, and/or an increased differentiation of osteoclasts from their precursor cells.

### 3.2. Osteoclast Differentiation in RA

Osteoclasts originate from the fusion of mononucleated cells belonging to the myeloid hematopoietic lineage in the presence of macrophage colony-stimulating factor (M-CSF) and receptor activator of nuclear factor-κB ligand (RANKL). Osteoclasts were thought to arise only from common bone marrow myeloid progenitors [22] and peripheral monocytes [23] under inflammatory conditions until it was shown in vitro that conventional human and murine immature dendritic cells were also able to transdifferentiate into osteoclasts [24,25,26]. This confirms the high cellular plasticity within the myeloid lineage and establishes conventional dendritic cells as an additional osteoclastic precursor. In vitro transdifferentiation was later confirmed in vivo by the transfer of dendritic cells into nonobese diabetic/severe combined immunodeficient (NOD/SCID) mice following co-culture with activated CD4+ T-lymphocytes [27], in oc/oc osteopetrotic mice deficient in a subunit of the vacuolar ATPase [28], and later in a model of aseptic bone inflammation [29]. Therefore, it is likely that osteoclast arises from a mix of various osteoclast precursor cells from bone marrow progenitors to cells already engaged in the monocyte or dendritic cell pathways depending on the signals they receive from their environment (Figure 2).

Osteoclast differentiation is orchestrated by the molecular triad RANKL, receptor activator of nuclear factor-κB (RANK), and osteoprotegerin (OPG), while RANKL, a TNF superfamily molecule, was originally cloned in T-cells [30], its function as the master regulator of osteoclastogenesis was not realized until the RANKL-knockout mouse was generated and mutations were identified in human osteopetrotic patients [16,28,29]. The binding of RANKL to its specific receptor RANK on mononuclear osteoclast precursors initiates a cascade of transcriptional changes culminating in osteoclast differentiation (Figure 3). OPG is a soluble decoy RANKL receptor that suppresses the RANKL function by competitively inhibiting RANKL-RANK binding. In response to RANKL stimulation, RANK recruits the adaptor molecule, TNF receptor-associated factor-6 (TRAF6), which subsequently activates downstream signaling pathways such as mitogen-activated protein kinases (MAPKs) and the transcription factor nuclear factor-κB (NF-κB). The activated NF-κB induces the nuclear factor of activated T-cells cytoplasmic 1 (NFATc1), the master transcription factor of osteoclast differentiation, and interestingly a key regulator of immune response. NFATc1 then translocates into the nucleus where it induces numerous osteoclast-specific target genes which are responsible for the osteoclast function. RANKL is highly expressed in the synovial tissue from RA patients [30,31,32,33]. The cells expressing RANKL in RA are not only osteoblasts that support osteoclastogenesis in physiology but also activated T-lymphocytes and mainly synovial fibroblasts [34]. RANKL: OPG ratio may predict the progression of joint destruction in RA patients. For example, a cohort study demonstrated that the RANKL: OPG ratio predicted annual radiological damage over 11 years [35].

Blockade of osteoclast differentiation by inhibiting RANKL has also demonstrated some efficacy in impairing the progression of bone erosion in arthritic mice [17,18,19,20,21,36]. Two phase 2 studies using a RANKL-specific monoclonal blocking antibody (denosumab) have been conducted in patients with RA receiving methotrexate and both demonstrated significant inhibition of bone erosion progression compared to the placebo without the inhibition of inflammation itself [37,38]. Recently, in the DESIRABLE study, a placebo-controlled phase 3 trial, denosumab significantly inhibited the progression of joint destruction and increased lumbar spine bone mineral density (BMD) in patients receiving concomitant conventional DMARD [39]. However, so far anti-resorptive drugs targeting osteoclasts such as denosumab as well as bisphosphonates are inadequate since they also alter physiological bone remodeling, necessitating the discovery of new targets. Interestingly, we have recently identified autotaxin as a such promising drug target candidate. Autotaxin (ATX), also known as nucleotide pyrophosphatase-phosphodiesterase 2 (NPP2) is a lysophospholipase D highly expressed both in synovium from RA patients and arthritic animal models [40] and is responsible for the cleavage of lysophosphatidylcholine (LPC) to lysophosphatidic acid (LPA), a serum-borne factor mandatory in vitro for RANKL-induced osteoclast formation [41,42]. We have shown that the inhibition of ATX prevents focal and systemic bone resorption in inflammatory animal models without affecting inflammation. Remarkably, physiological bone remodeling and non-inflammatory-induced bone loss are unaffected (Figure 2 and Figure 3).

### 3.3. Triggers for Bone Resorption in RA

#### 3.3.1. Innate Immune Mechanisms

Accumulating evidence suggests a role for innate immune mechanisms in enhanced osteoclastogenesis and bone loss in RA. Osteoclasts express innate immune receptors such as immunoreceptor tyrosine-based activation motif (ITAM)-associated receptors and Toll-like receptors (TLRs).

As other immune cells, osteoclasts require co-stimulatory signals for their activation in addition to the signaling pathway induced by RANKL. ITAM-harboring proteins, DNAX-activating protein of 12 kDa (DAP12), and Fc gamma receptor (FcRγ), associate with surface receptors, triggering receptor expressed in myeloid cells 2 (TREM2) and osteoclast-associated receptor (OSCAR), respectively. However, the ligands activating the receptors in osteoclast progenitors are not known. DAP12 and FcRγ co-stimulatory signals are required for osteoclastogenesis through the activation of spleen tyrosine kinase (SYK) and phospholipase C-gamma 2 (PLC-γ2) downstream signaling, regulating the calcium-calcineurin-NFATc1 pathway [43,44] (Figure 3). The ITAM signaling pathway is activated in RA. In fact, it has been shown that OSCAR was expressed by osteoclasts at the sites of bone erosions in the joints of RA patients and the percentage of cells expressing OSCAR, FcRγ, DAP12, and TREM2 was significantly higher in RA synovium compared to healthy synovium [45,46]. These two pathways, RANK and ITAM, cooperatively stimulate the activation of NFATc1. Takayanagi’s group has shown that the activation of tyrosine kinase Btk by RANK and BLNK (B-cell linker, phosphorylated adapter by Syk) by receptors with the ITAM domain allows the formation of the Btk/BLNK complex. This complex induces the phosphorylation of PLC-γ resulting in a calcium-dependent activation of NFATc1 and thus stimulates osteoclastogenesis [47]. The inhibition of Btk is a promising strategy for the prevention of bone loss in osteoclast-associated bone disorders such as RA [48].

The expression of TLRs is highly upregulated in various cells during joint inflammation in RA [49,50], whereas TLR stimulation induces the expression of RANKL in synoviocytes and osteoblasts, thereby favoring osteoclastogenesis and bone erosion. In osteoclasts, TLR ligands may differentially impact osteoclastogenesis and osteoclast survival depending on the osteoclast maturation stage [50,51]. The role of TLRs in bone loss in RA thus remains to be fully elucidated.

#### 3.3.2. T-Cells

Activated T-cells, in particular T-helper (Th) 17 and regulatory T-cells (Treg), have emerged as primary players in RA pathogenesis of bone loss. Th17 cells are considered to be the most osteoclastogenesis-inducing T-cells. Th17 cells produce interleukin-17 (IL-17), which in turn induces RANKL expression by synovial fibroblasts, the major source of RANKL in RA, as well as RANK expression on the osteoclast precursor [52,53] (Figure 2). IL-17 promotes osteoclastogenesis, bone resorption, and production of cytokines TNF and IL-1 by macrophages and IL-6 by fibroblasts [54,55]. Conversely, its inhibition leads to the improvement of inflammatory arthritis animal models [41,42]. Nevertheless, IL-17 inhibition has demonstrated only a limited effect in the treatment of active RA [56,57,58,59].

The IL-17-producing Th17 cells are the exclusive osteoclastogenic T-cell subset as Th1 and Th2 subsets inhibit osteoclastogenesis through their respective canonical cytokines IFN-γ and IL-4 [52,60]. Treg cells also inhibit osteoclastogenesis, indirectly through anti-inflammatory cytokines such as IL-10 and directly through cell contact and cytotoxic T-cell lymphocyte antigen 4 (CTLA4), a negative regulatory costimulatory molecule expressed in particular on Treg cells [61,62,63]. CTLA-4 competes with CD28 for binding to CD80 and CD86 molecules expressed on osteoclast precursors, as well as on other antigen-presenting cells. In addition to preventing persistent and pathological T-cell activation, CTLA-4 arrests osteoclastogenesis, even in the presence of RANKL (Figure 3). The anti-erosive effect of abatacept, a CTLA4-Ig fusion protein efficacious in patients with RA, underlines this effect and again the immune-bone interactions. Deficiencies in Treg cell function and Th17/Treg cell imbalance have been identified in RA. Indeed, Treg cells derived from patients with active RA are defective in their ability to suppress cytokine production due to abnormalities in the expression and function of CTLA-4 [64]. However, data on the presence and distribution of Treg cells in inflamed synovial tissue and on the effects of abatacept on Treg cell function are both limited and conflicting [65,66,67,68] and further studies are needed.

#### 3.3.3. Pro-Inflammatory Cytokines

The correlation between bone loss and clinical disease activity, as well as the protection from bone erosion progression by a tight therapeutic control of synovitis, support the concept that chronic inflammation is the major mechanism involved in bone loss in RA [69].

Pro-inflammatory cytokines, including TNF, IL-6, and IL-1, play an important role in perpetuating inflammatory and destructive processes in RA. TNF is a key pro-inflammatory cytokine among them as its overexpression is sufficient to induce arthritis in mice (hTNFtg) [70]. TNF stimulates osteoclastogenesis both indirectly and directly. Firstly, TNF upregulates RANK and c-fms (M-CSF receptor) expression in osteoclast precursors and RANKL expression in synovial fibroblasts in conjunction with IL-6 [71,72]. Secondly, TNF directly stimulates the differentiation of osteoclasts from mononuclear precursors in synovial tissues in synergy with RANKL [73] (Figure 2). TNF also expands the pool of osteoclast precursors and induces the production of other pro-inflammatory cytokines [74]. 

TNF also stimulates the connective tissue growth factor (CTGF) production by synovial fibroblasts in patients with RA [75]. CTGF, also known as CCN family member 2 (CCN2), is a bioactive cytokine believed to be a downstream mediator of transforming growth factor (TGF)-β. CTGF promotes osteoclastogenesis and aberrant osteoclastogenesis observed in collagen-induced arthritis (CIA) mice is reduced by neutralizing anti-CTGF monoclonal antibody [75,76].

IL-6 is produced by macrophages and synovial fibroblasts. IL-6 promotes osteoclastogenesis by upregulating RANKL expression in synoviocytes and is also an important factor for Th17 differentiation [77]. Similarly to TNF, IL-1 is produced by monocytes/macrophages and promotes osteoclast differentiation synergistically with TNF [78]. In addition, IL-1 directly increases the ability of osteoclasts to resorb [79]. As a consequence, treatments targeting pro-inflammatory cytokines have been shown to retard or arrest the occurrence of bone erosion and improve skeletal remodeling [80,81,82]. In addition, other cytokines such as IL-15, IL-33, IL-34 promote osteoclastogenis and may represent a new biologic therapeutic target for RA. 

#### 3.3.4. Autoimmunity

Due to the fact that chronic inflammation is considered as the primary trigger for bone damage, the discovery of systemic bone loss and bone erosion in healthy ACPA-positive individuals suggests that bone loss and bone resorption precede the clinical onset of inflammation and may be caused not only by inflammation, absent during the pre-clinical phases of RA, but also by autoantibodies such as ACPAs, present long before arthritis [7,83,84,85]. In accordance with this hypothesis of a direct pathogenic role of ACPAs in bone loss, the injection of ACPAs isolated from RA patients into immunodeficient mice induces osteopenia and enhanced osteoclast number [86].

Direct and indirect mechanisms of ACPA-mediated bone loss have been suggested. Due to the fact that ACPAs mainly belong to the IgG subtype, they are recognized by FcγR in the form of immune complexes with their citrullinated antigens (Figure 3). It was originally proposed that ACPA-containing immune complexes bind to FcγR on immune cells such as monocytes/macrophages resulting in the release of TNF, which in turn stimulates osteoclastogenesis [87,88,89]. In addition, as osteoclast precursors belong to the myeloid cell lineage, they also express the FcγR and can therefore bind immune complexes. Crosslinking of these FcγRs directly promotes osteoclastogenesis. Accordingly, conditional knockout mice lacking FcγR IV in osteoclasts are partly protected from bone loss in an arthritis model [90]. In recent years, studies have shown that ACPAs bind to citrullinated proteins such as vimentin on the surface of osteoclast precursors and directly stimulate their differentiation into mature osteoclasts [86,91] (Figure 2 and Figure 3), but some have been retracted or corrected [91,92,93,94] due to errors in methodology and further information remains sparse. In addition, osteoclast precursors highly express enzymes performing protein citrullination (peptidylarginine deiminases, PAD) and that citrullination of vimentin in osteoclast precursors is increased in RA [86].

Alterations in the glycosylation of serum IgG in RA have long been noted [95]. A lack of total IgG Fc glycosylation correlates strongly with disease activity and severity, and reverses to normal levels upon effective treatment [95,96,97,98,99]. Glycosylation of the IgG Fc domain influences the binding of IgG to FcγR and determines the engagement of both pro- and anti-inflammatory FcγR. Removal of the Fc glycans diminishes the IgG Fc-mediated biological activity due to the failure of the non-glycosylated molecule to bind to FcγR [100]. Furthermore, only desialylated immune complexes enhance osteoclastogenesis in vitro and in vivo. Indeed, the administration of a sialic acid precursor results in increased IgG sialylation and decreased bone erosion in mice with CIA [101]. Remarkably, ACPA levels increase in the pre-clinical phase of RA, but ACPA glycosylation profile changes towards a pro-inflammatory profile only within the 3 months prior to the clinical onset of RA. This shift is regulated by Th17 cells in an IL-22- and IL-21-dependent manner [101,102].

Insofar as ACPA can promote bone loss and some DMARDs can decrease ACPA levels in RA patients, the goal of achieving immunological remission with the treatment is enticing [103]. However, the real value of reducing ACPA in RA patients still needs to be determined, as well as the mechanisms regarding bone loss in ACPA-negative RA patients. An alternative mechanism might involve other posttranslational modifications of proteins such as carbamylation. Indeed, similarly to ACPAs, anti-carbamylated protein antibodies are also a strong predictive factor for the development of structural damage in RA [104,105,106]. However, their pathogenic role in osteoclastogenesis and bone loss has not yet been studied.

#### 3.3.5. MicroRNAs

MicroRNAs (miRNAs or miRs) are small non-coding single-stranded RNAs that play an important role in the epigenetic regulation of gene expression by binding in a complementary manner to messenger RNAs and therefore inhibiting the expression of their target gene at the posttranscriptional level. They target more than 60% of the genome as each miRNA can regulate the expression of multiple target genes and a single gene expression can be regulated by multiple miRNAs in a synergistic or opposing way. 

MiRNAs have recently emerged as critical regulators in the pathogenesis of RA by regulating in particular inflammation and osteoclastogenesis. Further elucidation of pathways regulated by miRNAs in RA may provide new targets for therapy. For example, the expression of miR-155 is upregulated in synovial fibroblasts and macrophages from patients with RA [107]. MiR-155-deficient mice are protected from CIA and show a significant reduction in osteoclast infiltration and bone erosions, without affecting joint inflammation in the K/BxN serum-transfer induced arthritis model [108]. Conversely, the expression of miR-17-5p is reduced in synovial fibroblasts and Najm et al. have demonstrated an anti-inflammatory and anti-erosive role of miR-17 in the CIA mouse model [109]. Similarly, it has been shown that the administration of miR-124 in rats with antigen-induced arthritis results in the amelioration of arthritis, downregulation of NFATc1 and RANKL expression, inhibition of osteoclastogenesis, and attenuation of bone destruction [110]. 

#### 3.3.6. Autophagy

Autophagy consists of lysosome-mediated degradation of dysfonctional cellular components such as organelles in response to various cellular stress conditions such as inflammation [111]. Several studies have described that autophagy and autophagy-related protein activation are associated with osteoclast biology. TNF regulates osteoclast differentiation and activation via stimulation of the autophagy pathway [112]. The knockdown of p62, the most specific adaptor protein associated with autophagy, inhibits autophagy activation in osteoclasts induced through RANKL and also inhibits osteoclastogenesis [113]. Autophagy may act as a novel mediator of the destruction of articular bone in RA patients [114]. However, recent findings show that autophagy may play dual roles in the regulation of bone resorption. Indeed, optineurin (OPTN) is an autophagy adaptor/receptor that regulates NF-κB signaling. Lee et al. demonstrated its protective role against joint destruction in RA acting as a negative regulator of RANKL [115].

## 4. Inhibition of Osteoformation in RA

Bone loss in RA is the result of exacerbated osteoclast mediated bone resorption and blunted bone formation. In marked contrast with the physiological situation, bone resorption in the high inflammatory context is not compensated by bone formation. Treatments using anti-RANKL, anti-TNF, or anti-IL-6 receptor aiming at decreasing the high level of these pro-osteoclastogenic cytokines, do not reverse bone formation inhibition in RA patients under therapy [116]. Furthermore, the current intermittent parathyroid hormone (PTH) anabolic treatment combined with TNF blockers fail to reduce the erosion volume in patients with established RA but with the controlled disease activity [117], reflecting a total uncoupling between bone resorption and bone formation, which is still not fully understood. By contrast to humans, the treatment of hTNFtg mice with a combined therapy consisting of anti-TNF together with intermittent PTH led to the regression of local bone erosion and bone repair, demonstrating new bone formation [36].

Bone development and bone formation rely on Wingless (Wnt) signaling, which is required in osteoblastic lineage differentiation and function. The Wnt pathway controls the different steps of bone forming osteoblast differentiation from the mesenchymal stem cells (MSCs) commitment to the osteoblast progenitor, the amplification of pre-osteoblasts, final osteoblast differentiation, and apoptosis [118]. To prevent bone over-growth, Wnt pro-osteogenic functions are antagonized by a set of physiological factors interfering with the canonical Wnt receptors frizzle and LPR5 or LPR6 [119]. In RA, activated synoviocytes produce high levels of RANKL and TNF which in turn induce, in an autocrine loop, over expression of Wnt antagonists namely Dickkopf proteins DKK-1 [120], -2, and sclerostin [121] leading to bone formation inhibition (Figure 2).

The circulating DKK-1 level is increased in RA patients’ sera and correlates with the inflammation and erosion grade, as such it could serve as a biomarker of disease activity [122]. The DKK-1 level is also elevated in arthritic models such as CIA and hTNFtg mice. Therefore, targeting Wnt antagonists to improve bone formation and repair, may offer an alternative to combined therapy [5]. However, the surprisingly enough treatment with anti-DKK1 do not improve bone formation except in hTNFtg mice [122].

Sclerostin is also an attractive therapeutic target for bone loss pathologies. Sclerostin-neutralizing antibodies have a strong bone-building effect in mice, rats, monkeys, and humans [123,124,125]. In CIA mice, sclerostin neutralizing antibodies administration prevents BMD decrease and that of the bone volume at axial and appendicular sites but does not protect from erosion on the periarticular bone and fails to repair focal erosions [126]. However, in the hTNFtg mouse model, Wehmeyer et al. showed that TNF upregulates sclerostin production in inflammatory synoviocytes but when they generated hTNFtg/sclerostin-deficient (Sost^−/−^) mice, they reported, in the absence of sclerostin, an exacerbation of the disease rather than the expected bone repair and reversal of inflammatory bone erosions [121]. These observations suggest that sclerostin may have further functions in addition to its major role in the Wnt pathway, or may act as an anti-osteoclastogenic factor in the TNF-dependent arthritis model. To support this hypothesis of potential uncovered sclerostin functions or regulation, recent findings show that when murine skeletal stem cells over-expressing sclerostin are in vivo engrafted, they can generate overgrown bones against all odds [127]. Though, further studies are needed using the Sost tissue-specific ablation in arthritic animal models, to get a better understanding of the precise role of sclerostin in RA.

MiRNAs have been shown to play a regulatory role not only in bone resorption, but also in bone formation. Using the serum transfer mouse model of RA, Iwamato et al. identified 22 miRNAs whose expression in the synovial tissue significantly differed between arthritic and nonarthritic mice and several of which targeted the Wnt signaling pathway. Indeed, miR-221-3p was upregulated by synovial fibroblasts treated with TNF and its overexpression suppressed osteoblast differentiation and mineralization, suggesting that miRNAs derived from inflamed synovial tissues may regulate signaling pathways at erosion sites that affect bone loss and potentially also compensatory bone formation [128]. Conversely, miR-218 expression is altered during the osteogenic differentiation of synovial fibroblasts from RA patients and miR-218 promotes this osteogenic differentiation process by suppressing DKK-1 [129].

The inhibition of osteoblast bone formation in inflammatory conditions may not be only secondary to RANKL and TNF induction of Wnt pathway antagonists. Chen et al. reported that RANKL signaling directly inhibits the osteogenesis of bone marrow MSCs (BMSCs), which express RANK, its cognate receptor at early stages. The specific deletion of RANK in BMSCs with Prx1-Cre mice leads to a higher bone mass phenotype [130]. B-memory cells from RA patients or from RA mice models also express RANKL [131].

At bone erosion sites, inflammation leads to an osteoimmune focus of multiple activated cell types producing a locally high level of RANKL and TNF, including inflammatory synoviocytes, Th17 cells, and B-cells. B-cells are enriched at the subchondral and endosteal bone marrow areas in direct contact with the bone surface and osteoblasts. RA B-cells produce high levels of TNF and CCL3 (chemokine ligand3) that directly inhibit bone marrow MSCs osteogenic differentiation through NF-κb and Erk activation [132]. At the inflamed joint, the local source of cytokines which are both pro-osteoclastogenic and anti-osteogenic, team up to achieve bone destruction and bone formation inhibition rendering anabolic treatment inefficacious in the absence of a pro-osteogenic cytokine as IL-22 in the RA disease. 

New therapeutic strategies are being developed based on the immunomodulatory properties that MSCs acquired under inflammatory conditions. MSCs from bone marrow or from the umbilical cord blood produce soluble factors including transforming growth factor β1 (TGF- β1), prostaglandin E2 (PGE2), hepatocyte growth factor (HGF), and IL-10. These immunosuppressive factors affect B- and T-cell proliferation and differentiation, dendritic cell maturation, and finally natural killer cell activity. A substantial number of in vitro or pre-clinical studies reported that the pulsed electromagnetic field (PEMF) stimulates the MSCs chondrogenic and osteogenic potential, leading to an increased differentiation and matrix deposition. The PEMF treatment also amplifies MSCs immunoregulatory functions resulting in a decay of inflammatory cytokines including TNF, IL-1β, and IL-6 (for details see [133]). Although the RA treatment by the administration of sequential PEMF exposure seems attractive, non-invasive, and riskless, this strategy has to be evaluated in new clinical studies for its beneficial efficacy over time, as up to date, very few have been reported [134]. In vivo MSCs infusion, induce peripheral tolerance and have been used as the preventive treatment for graft versus the host disease (GvHD) [135,136]. Therefore, MSCs cell-based therapies can be envisioned to contain the local release of inflammatory cytokines by the injured tissue. A phase I clinical trial has been performed in RA patients with moderate disease activity, treated by a single intravenous injection of umbilical cord blood-derived MSCs. This trial revealed no short-term safety concerns and within 24 h post treatment a substantial reduction of IL-1β, IL-6, IL-8, and TNF, as well as an increase in the IL-10 content. However, a long term evaluation study using a larger cohort is needed to consider this therapeutic strategy [137]. 

Beyond the crosstalk between the immune and skeletal systems, adipocytes also regulate, through the release of adipokines, bone remodeling in physiological and pathological conditions such as RA, expanding the field of osteoimmunology to osteoimmunoendocrinology. 

## 5. Conclusions

Bone loss is a hallmark of RA and a severe outcome as it deteriorates the functional capacity of the patients. Whereas, it is well established that chronic inflammation represents the major trigger for bone loss in RA supporting the current therapeutic strategy of targeting the best control of local and systemic inflammation, the substantial progress made in understanding interactions between the immune system and bone gradually yield to a paradigm shift. Firstly, bone loss commences early in the course of RA even before the onset of inflammation during the pre-clinical phase of the disease and may be also triggered by autoimmune and innate immune mechanisms. Secondly, it is now evident that inflammation also blunts osteoblast differentiation and function in RA. Therefore, a better understanding of the molecular mechanisms involved in pre-clinical bone loss and osteoformation will provide potential new therapeutic approaches to prevent and repair bone loss, respectively.

## Figures and Tables

**Figure 1 biomolecules-11-00048-f001:**
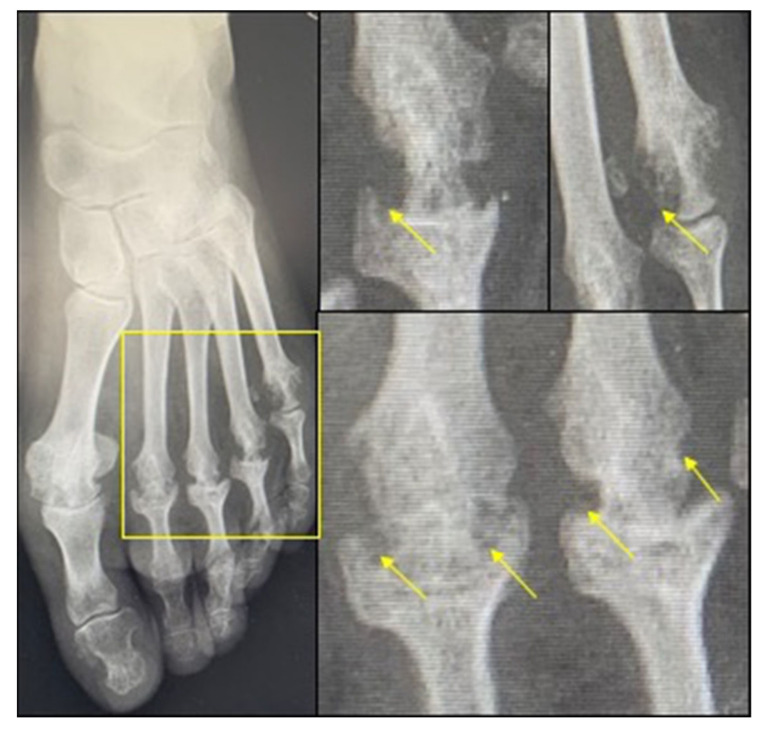
Bone erosions of the foot in rheumatoid arthritis (RA). Arrows show erosions on metacarpophalangeal joints.

**Figure 2 biomolecules-11-00048-f002:**
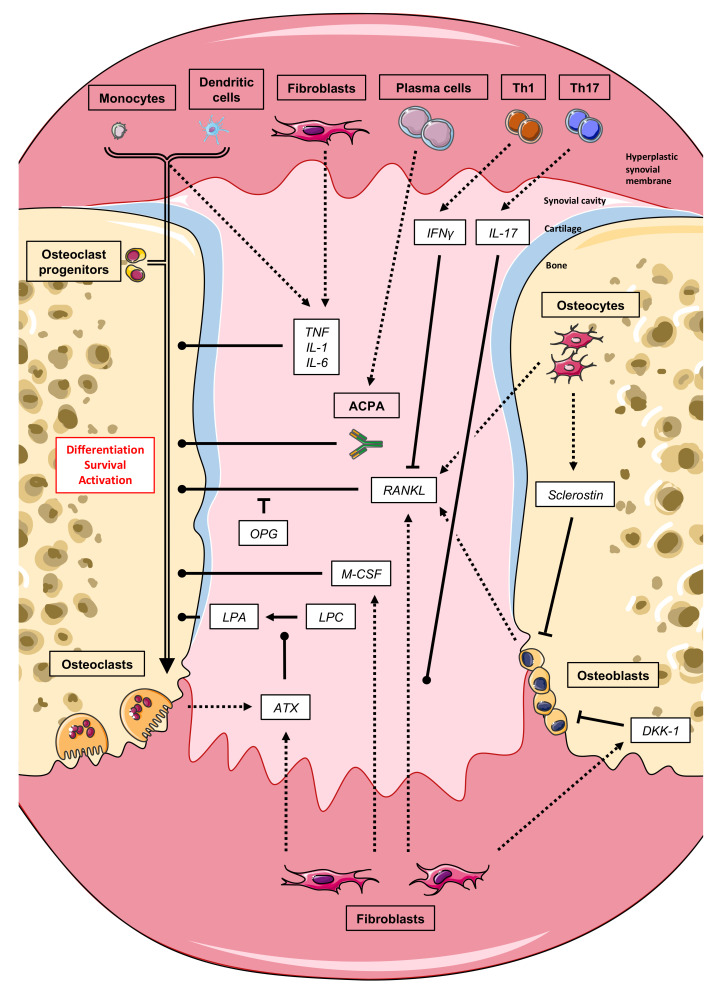
Synovial membrane and bone interactions in rheumatoid arthritis. The inflamed synovium produces inflammatory mediators that enhance osteoclastogenesis, mostly by the upregulation of the receptor activator of NF-κB ligand (RANKL), leading to articular bone erosions development. In addition, osteoclasts as well as their precursors can be directly activated by anti-citrullinated protein antibodies (ACPAs) present long before during the pre-clinical phase of the disease. Osteoblast differentiation, on the other hand, is inhibited by antagonists of the Wnt signaling pathway, including Dickkopf (DKK-1) and sclerostin, preventing erosion repair. ACPA: Anti-citrullinated peptide antibodies; ATX: Autotaxin; DKK-1: Dickkopf related protein 1; IFN: Interferon; IL: Interleukine; LPA: Lysophosphatidic acid; LPC: Lysophosphatidylcholine; M-CSF: Macrophage colony-stimulating factor; OPG: Osteoprotegerin; RANK: Receptor activator of nuclear factor kappa B; RANKL: RANK ligand; TNF: Tumor necrosis factor; Th: T-cell; dashed triangle arrowheads: Secretion; simple triangle arrowheads: Chemical reaction; round arrowheads: Activation; flat arrowheads: Inhibition.

**Figure 3 biomolecules-11-00048-f003:**
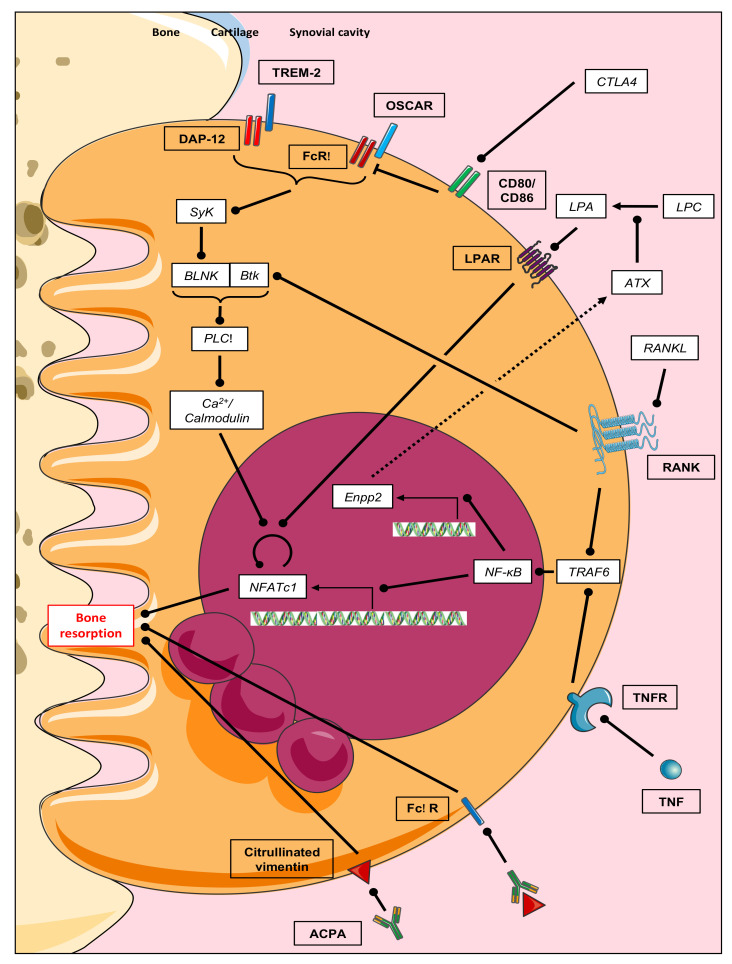
Enhanced osteoclast bone resorption in rheumatoid arthritis. Bone resorption is mediated by multiple pathways (RANKL/RANK, inflammatory cytokines, ITAM/Ca^2+^, and ATX/LPA) converging on NFATc1. Additionally, bone loss is mediated by direct and indirect ACPA-related mechanisms. ACPA: Anti-citrullinated peptide antibodies; ATX: Autotaxin; BLNK: B-cell linker protein; BtK: Burton tyrosine kinase; CD: Cluster of differentiation; CTLA4: Cytotoxic T-lymphocyte-associated protein 4; DAP-12: DNAX-activating protein of 12 kDa; Enpp2: Ectonucleotid pyrophosphatase/phosphodiesterase 2; FcγR: Fc gamma receptor; FcRγ: Fc receptor gamma subunit; IgG: Immunoglobulin G; IL: Interleukine; LPA: Lysophosphatidic acid; LPAR: LPA receptor; LPC: Lysophosphatidylcholine; MHC-1: Major histocompatibility complex 1; NFATc1: Nuclear factor of activated T-cells cytoplasmic 1; NF-κb: Nuclear factor kappa B; OSCAR: Osteoclast associated receptor; PLCγ: Phospholipase C gamma; RANK: Receptor activator of nuclear factor kappa B; RANKL: RANK ligand; Syk: Spleen tyrosine kinase; TNF: Tumor necrosis factor; TRAF6: TNF receptor-associated factor 6; TREM-2: Triggering receptor expressed on myeloid cell. 2. Dashed triangle arrowheads: Secretion; simple triangle arrowheads: Chemical reaction; round arrowheads: Activation; flat arrowheads: Inhibition.

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
