# Peer review of "Rheumatoid Arthritis in the View of Osteoimmunology"

_biomolecules, 2020, doi:10.3390/biom11010048_

Round 1
Reviewer 1 Report
This review is current and comprehensive regarding bone reactions in rheumatoid arthritis but is quite inaccurate with respect to origin and evolution of erosions at the periphery of rheumatoid joints.
Joint erosions of rheumatoid arthritis start by devitalization of cartilage (both matrix and chondrocytes) at the periphery of the joint by the actions of inflammatory molecules elaborated by neutrophils and cells within the synovium, principally macrophages. This leads to ingress of capillary blood vesels into the degraded marginal cartilage and expansion of hyperplastic synovium onto the cartilage surface, pannus.
Degradation of cartilage matrix proteins by proteases generated in the inflammatory response results not only in Anti citrullinated protein antibody but also many other protein antibodies that are less well characterized with respect to their immunologic functions.
As the process proceeds, the cartilage is resorbed by chondroclasts , multinucleated cells from the same linage as osteoclasts. Only when the bone is exposed and the matrix on the surface of the bone is degraded do the processes described in the paper occur.
Therefore, Figure 2 is quite misleading.
Further the location of bone resorption and its extent are likely determined by biomechanical and physiologic factors distinct from inflammatory and immunologic mechanisms.
it is suggested that the relationships of osteoimunology to rheumatoid arthritis, important as they are, will be understood well only when placed in an accurate context of rheumatoid cartilage and bone pathophysiology.
Reviewer 2 Report
Authors presents an interesting and well-organized review. The manuscript highlights the importance of osteoimmunology as a key relationship between immune and bone systems to better understand the pathophysiology of systemic and local bone loss in rheumatoid arthritis. However, there are some concerns that need to be addressed.
- Taken into account the aim to show and summarize the substantial progress in osteoimmunology and bone loss in rheumatoid arthritis, presented references have to be updated, less than 10% of cited references were published in the last 3 years.
- Some aspects should also be approached. For instance, autophagy (1002/art.41290), chondrocytes or MSCs as other pivotal cells (10.3389/fimmu.2019.00266), microRNA (10.3390/ijms20205141), adipokines (10.3390/ijms20174091), or other factors like ctgf (10.1186/ar2863).
- Figure 2. Self-explanatory caption should be added.
- Definition of NOD/SCID mice or sost-/- mice is recommendable.
- Correct mistake. 3.3.4. Autoimmunity instead of 3.3.3. Autoimmunity
Round 2
Reviewer 1 Report
That the purpose of this paper is to review the osteoimmunology of rheumatoid arthritis is accepted.The concept that subclinical inflammation usually affects bone early in disease is controversial, but stating this idea within the review is also acceptable.
However, Figure 2,even recognizing this is a schematic, shows advanced disease with cartilageand bone erosion..
Figure 2 is erroneous because while the bone is shown to be actively eroding, the cartilage, synovium interface appears unaffected by disease. Whenever bone is morphologically involved, the synovium is hyperplastic and cartilage at the interface shows erosion..
As well, the erosion and new bone formation are shown on the surface of the bone; the underlying bone appears normal. This is incorrect. The subchondral bone and bone marrow are profoundly affected and are in fact the focal location of the osteoimmunologic interactions
